# Experience drives innovation of new migration patterns of whooping cranes in response to global change

Claire S. Teitelbaum[1,2], Sarah J. Converse[3], William F. Fagan[4], Katrin Böhning-Gaese[1,2], Robert B. O'Hara[1], Anne E. Lacy[5] & Thomas Mueller[1,2]

Anthropogenic changes in climate and land use are driving changes in migration patterns of birds worldwide. Spatial changes in migration have been related to long-term temperature trends, but the intrinsic mechanisms by which migratory species adapt to environmental change remain largely unexplored. We show that, for a long-lived social species, older birds with more experience are critical for innovating new migration behaviours. Groups containing older, more experienced individuals establish new overwintering sites closer to the breeding grounds, leading to a rapid population-level shift in migration patterns. Furthermore, these new overwintering sites are in areas where changes in climate have increased temperatures and where food availability from agriculture is high, creating favourable conditions for overwintering. Our results reveal that the age structure of populations is critical for the behavioural mechanisms that allow species to adapt to global change, particularly for long-lived animals, where changes in behaviour can occur faster than evolution.

[1] Senckenberg Biodiversity and Climate Research Centre, Senckenberg Gesellschaft für Naturforschung, Senckenberganlage 25, 60325 Frankfurt, Germany. [2] Department of Biological Sciences, Goethe University, Max-von-Laue-Straße 9, 60438 Frankfurt, Germany. [3] US Geological Survey, Patuxent Wildlife Research Center, 12100 Beech Forest Road, Laurel, Maryland 20708, USA. [4] Department of Biology, University of Maryland, College Park, Maryland 20742, USA. [5] International Crane Foundation, P.O. Box 447, E-11376 Shady Lane Road, Baraboo, Wisconsin 53913, USA. Correspondence and requests for materials should be addressed to C.S.T. (email: claire.teitelbaum@gmail.com) or to T.M. (email: thomas.mueller@senckenberg.de).

Animal migration patterns are shifting rapidly as changes in both climate and land use modify historical habitats and alter environmental conditions[1–3]. Winter shortstopping, the shortening of migration routes by shifting wintering grounds towards the breeding grounds[1,2,4–6], has been documented in at least 27 bird species on four continents, and is likely to be even more widespread because data about birds' wintering grounds are often sparse[2,4]. Shortstopping can benefit migrants by decreasing the amount of energy they spend on long-distance flight[7] and facilitating earlier arrival on the breeding grounds[8], but these benefits require finding suitable new wintering sites closer to the breeding grounds. Changes in climate and land use can create suitable new sites, where warmer temperatures decrease energy expenditure when overwintering at higher latitudes[7,9,10] and increased food availability from land use change (for example, land conversion to agriculture) can further offset the increased energetic requirements of overwintering in a colder place[4,11,12].

While many studies have investigated these extrinsic drivers of changes in migration patterns, the intrinsic mechanisms of shortstopping remain largely unknown[13]. Evidence indicates that genetic processes can lead to changes in migration via heritability or epigenetic regulation of migration behaviour[6,14,15]. However, migrations can change quickly, suggesting that behavioural mechanisms of adaptation, which produce changes within the lifespan of a single individual, may play a critical role in facilitating changes in migration patterns[2,3,7,13]. In long-lived bird species, individuals can improve their migratory performance over the course of their lifetime[16] and, in social species, individuals can alter their migration patterns based on the behaviour of other birds in the population[17–19]. These same experience-driven mechanisms may also contribute to population-level shifts in migratory behaviour and thus could be a major driver of shortstopping.

Here we investigate the rapid appearance of shortstopping behaviour in the reintroduced eastern migratory population of whooping cranes (*Grus americana*). Since 2001, over a century after migratory whooping cranes were extirpated from eastern North America[20], captive-bred whooping cranes have been trained to migrate by following an ultralight aircraft from central Wisconsin (~ 43.9° N) to the Gulf Coast of Florida (~ 28.6° N). After their first ultralight-led migration, birds perform all subsequent migrations unaided. In addition, in recent years some juveniles have performed their first migration without ultralights, and instead have followed other birds in the population. Migratory movements of all individuals in the population have been extensively monitored since the beginning of the reintroduction, resulting in a unique dataset where birds in the population have been tracked throughout their lifetimes. We use these long-term monitoring data of 175 individuals to investigate both the intrinsic and extrinsic drivers of shortstopping in the population[21].

Before its extirpation, the population of whooping cranes breeding in North America's Upper Midwest probably over-wintered on the South Carolina coast[20], and likely possessed a migration comparable in distance (~1,500 km) to the current ultralight-trained migration. In the 14 years since reintroduction began, the mean migration distance of the population has decreased markedly from the 'trained' distance of almost 1,800 km to <900 km (Fig. 1a), with new overwintering sites as far north as Illinois and Indiana (~ 41.2° N). This population-level change resulted from changes in the behaviour of individual birds within their lifetimes (Fig. 1b), where some individuals used new sites, while others did not (Fig. 1b,c). The reduction in migration distance in the population, as compared to both the reintroduced and historic routes, required the establishment of new overwintering sites closer to the breeding grounds.

Since reintroduction began, the population has used 68 distinct overwintering sites, 65 of which were established independently of ultralight training.

To determine which intrinsic behavioural factors and extrinsic environmental drivers influence the establishment of these new northern overwintering sites, we quantified shortstopping by calculating the distance of each new overwintering site from the population's Wisconsin breeding grounds (43.87° N, − 89.23° E). We examined three factors hypothesized to explain the emergence of shortstopping. First, to test for an intrinsic behavioural mechanism of shortstopping, we examined the age of the oldest bird in the group that first used each overwintering site. In this population, the presence of older birds improves a group's migratory performance[15], which suggests that age could also drive the establishment of new sites.

In addition to this intrinsic driver, we tested the effects of environmental change in enabling shortstopping by examining two potential extrinsic drivers: grain cover and temperature anomaly. Corn cultivation is associated with shortstopping in sandhill cranes[22], and whooping cranes can feed on a variety of crops, largely grains[23], so we determined the percent grain cover within 10 km of each overwintering site as a measure of food availability from agriculture. Changes in climate could also drive shortstopping by decreasing the amount of energy required to overwinter at each site, so we calculated site-specific winter temperature anomalies. Temperature anomalies measure the difference in average winter temperature at each overwintering site between the estimated date of extirpation of the original migratory whooping crane population (ca. 1900) and today, and thus reflect the change in climatic suitability of each site over this period.

To assess the effects of these internal and external drivers on the emergence of shortstopping behaviour, we built a linear mixed-effects model[24,25] to measure the effects of group age, temperature anomaly and grain cover on site distance. We find that groups containing older, more experienced birds are the first to use shortstopping sites, and that these sites are in areas with relatively large increases in temperature and high grain cover, indicating that experience-based behavioural innovations can be major drivers of population-level responses to global change.

## Results

**Intrinsic drivers of shortstopping.** The age of the oldest bird in a migratory group was linked to the innovation of shortstopping, where groups with older birds were the first to use sites closer to the breeding grounds (Fig. 2a,d). Our model predicted that new sites were established 40 km closer to the breeding grounds for each additional year of age of the oldest bird in the group (Supplementary Table 1).

We also investigated whether these older birds were indeed more experienced with the new overwintering sites, which may have encouraged their innovation. Some individuals had previously stopped over at sites that they later established for overwintering (Fig. 1b), indicating that site familiarity may be important for the establishment of new sites. In addition, whooping cranes can recognize landscape features at relatively large spatial scales[26], meaning that an individual could have knowledge of the area surrounding a new overwintering site without having directly stopped over there. Thus, to measure the degree of familiarity of each individual with a new site, we identified the closest distance that each bird had been from a new site before establishing it for overwintering. We then tested whether this measure of experience was related to the age of each individual. Older birds were significantly more likely to be familiar with the areas surrounding the new overwintering sites

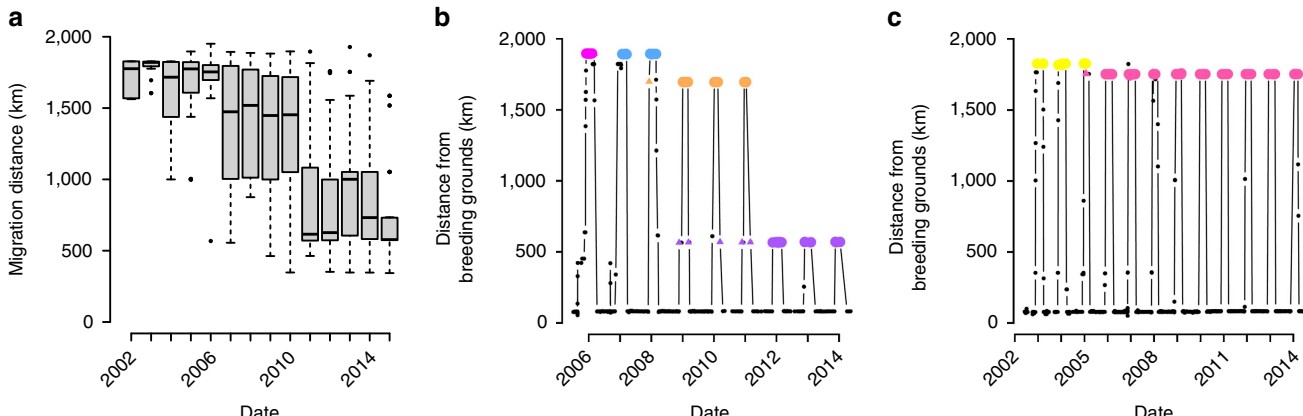

**Figure 1 | Emergence of shortstopping behaviour in whooping cranes.** (**a**) Shortstopping behaviour spread through the population. Boxplots show the median, interquartile range and Tukey-style whiskers of migration distance of all individuals in the population. (**b**) Relocations of one example individual that began shortstopping. Each colour represents a different overwintering site; smaller triangles in the same colour indicate stopover sites that were used for overwintering in later years (here orange and violet overwintering sites had been used previously as stopovers). (**c**) Relocations of a different example individual that exhibited relative consistency in overwintering behaviour.

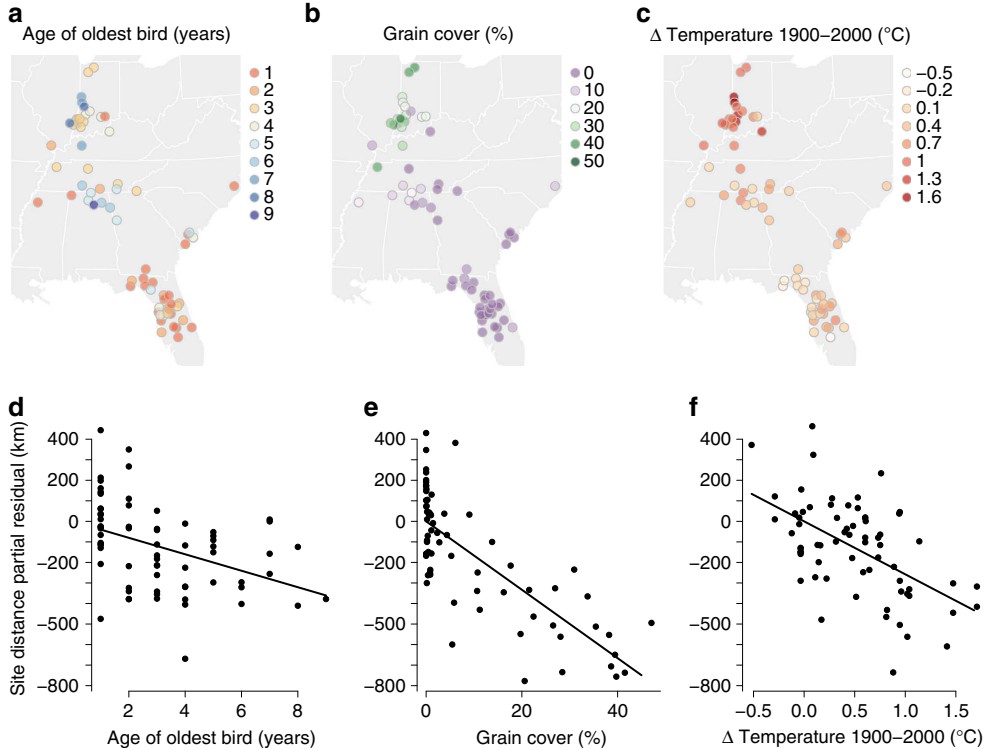

**Figure 2 | Locations of overwintering sites as related to intrinsic and extrinsic factors.** Upper panels **a**–**c** show all overwintering sites; lower panels **d**–**f** are partial residual plots, which display the effects of each predictor variable on the distance of each site from the breeding grounds, after accounting for all other effects in the model (Supplementary Table 1). Sites closer to the breeding grounds were first used by groups with older birds (**a,d**). Northern sites show higher grain cover than do southern sites (**b,e**). Northern sites have experienced relatively more warming since 1900 compared with southern sites (**c,f**).

than were younger birds; on average, birds over 6 years old had been within 19 km of a new overwintering site, whereas 1- and 2-year-old birds had been within 155 km (Fig. 3, Supplementary Table 2).

Because the average age of the population has increased since the beginning of reintroduction, we also performed randomization tests, which confirmed that the effect of age on the distance of a new site to the breeding grounds was independent of the year in which the site was first used (Supplementary Fig. 1).

**Social transmission of behaviour.** Whooping cranes migrate and overwinter in small but demographically heterogeneous groups[17,21], providing opportunities for social learning via interactions between birds of different ages. During the study period, whooping cranes overwintered in groups ranging from 1 to 22 individuals (mean $3.62 \pm$ s.d. 3.57 birds), with age ranges up to 11 years ($1.32 \pm 1.86$ years). Groups establishing new sites tended to be smaller ($2.32 \pm 1.37$ birds), but still had large age ranges ($2.83 \pm 2.93$ years). To investigate the spread of shortstopping through the population, we examined whether the

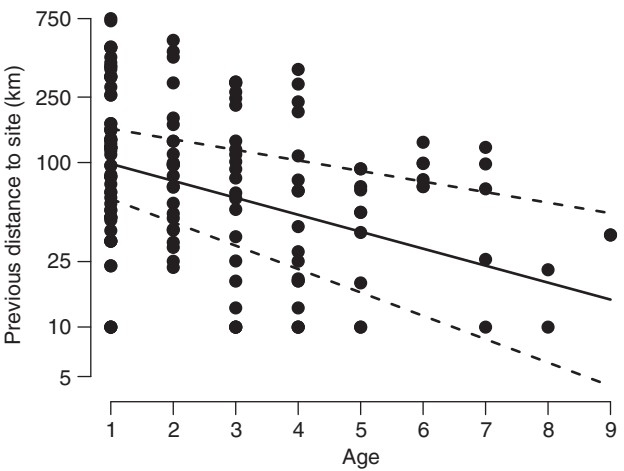

**Figure 3 | Older birds have been closer to future overwintering sites during previous migrations.** Distances between future overwintering sites and previously-used migration sites were shorter for older individuals than for young individuals, indicating that older birds may be more familiar with the areas surrounding new overwintering sites. Because data are known occurrences rather than complete tracks, the previous distance to each site represents a maximum possible distance for each individual. The line displays predictions from a linear mixed-effects model predicting previous distance to site from individual age and dashed lines show the 95% confidence interval of the mean (Supplementary Table 2).

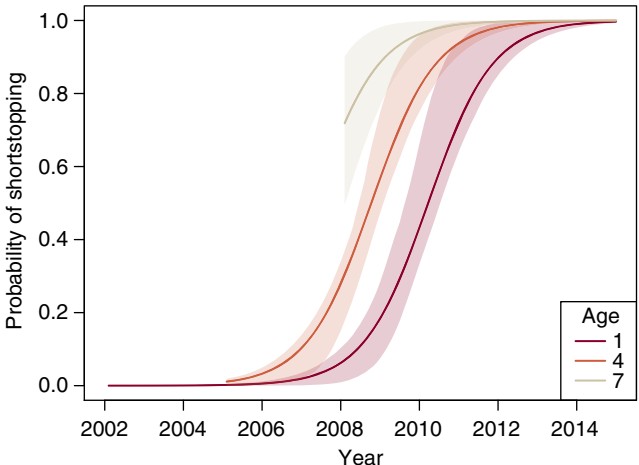

**Figure 4 | Older birds establish shortstopping behaviour.** The probability of shortstopping increases with both age and year, indicating that older birds initiate shortstopping but birds of all ages learn the behaviour over time. Curves represent predictions from a linear mixed-effects model testing the effects of age and year on the probability of shortstopping, including a random intercept of individual identity (Supplementary Table 3). Also shown are 95% confidence intervals of the predictions produced from 1,000 bootstrap replicates. Model predictions are plotted only for age and year combinations present in the data.

incidence of shortstopping changed over time in birds of a given age. We identified all birds that shortstopped in a given year (defined as migrating <1,200 km; Supplementary Fig. 2), then related the probability of shortstopping to year and individual age. The probability of shortstopping increased with both age and year (Fig. 4; Supplementary Table 3), indicating that older birds were the first to shortstop, but that birds of all ages eventually decreased their migration distance over the study period (Fig. 4). In 2006, no 1-year olds shortstopped, but by 2015, 75% of them did so.

**Extrinsic drivers of shortstopping.** The new, northern sites associated with shortstopping were in areas of high grain cover and more substantial climate change since 1900. Shortstopping sites closer to the breeding grounds had higher grain cover than sites near the original Florida overwintering grounds (Fig. 2b,e); the 10 sites closest to the breeding grounds had a mean grain cover of 28%, compared with 0% for the 10 sites farthest from the breeding grounds. The northernmost shortstopping sites were also in areas with more substantial climate warming since 1900 (Fig. 2c,f). Indeed, the 10 northernmost overwintering sites have warmed, on average, 0.93 °C more than the 10 south-ernmost overwintering sites.

## Discussion

Our results indicate that populations that include experienced individuals are more likely to be able to effectively and rapidly adapt behaviourally to environmental change, at least in long-lived, social species that exhibit social learning. Older individuals established behaviour that eventually spread to younger individuals in the population. The spread of novel behaviour via social learning is well known in birds[27], but, to our knowledge, has never before been shown to apply to colonization of new habitats or establishment of new migration patterns. Studies of animal cognition indicate that migratory species are less innovative than residents when it comes to foraging behaviour[28,29], but we show that migratory species are able to innovate new movement behaviours, possibly linked to their superior long-term memory[30] and ability to learn from experience.

The strong relationship between the age of the oldest individual in a group and the use of new, northern overwintering sites follows evidence from the studies of primate behaviour, where innovative behaviour often increases with age[31]. Older primates may be more innovative because they are more experienced and thus able to devise informed solutions to new problems[31]. Other animals with complex social structures, including elephants[32], cetaceans[33] and ungulates[34], live in groups led by older individuals, which has been attributed to the adaptive advantage of these older individuals' knowledge and experience[35]. In these social species, living in groups with older individuals can increase the reproductive success of the whole group, possibly via social transmission of learned behaviour[32]. In whales, post-reproductive females are more likely to lead group movement in years of low food availability, when using new movement paths is important for survival[33]. We show here that age and experience are important not only for group leadership and transmission of social information but also for innovating novel behaviours in a new or changing environment.

Differential migration by age is not uncommon in birds[16,36], including overwintering distributions where juveniles migrate farther than adults[37]. Age-related differences in migration distance can be due to dominance hierarchies, where older individuals use their higher social status to obtain better winter territories[38], and/or to reproductive pressure, where mature birds must arrive early on their breeding grounds to reproduce successfully[37]. However, these mechanisms are unlikely to apply to shortstopping, where there is a population-level shift in migration patterns in young as well as older birds. In whooping cranes in particular, the presence of young birds in groups that shortstop indicates that dominance does not determine migration distance (even though birds can be territorial during both winter and reproductive seasons[20]).

The newly established overwintering sites of the eastern migratory population of whooping cranes have considerably colder winter climates than do the historic overwintering locations of the species—the South Carolina coast, the Gulf

Coast and central Mexico[20]. Thus, although climate has warmed significantly since the extirpation of the original migratory population, temperature alone is not sufficient to explain shortstopping. Given that low temperatures are unlikely to be fatal for whooping cranes[39], the combination of temperature change and high food availability may facilitate overwintering farther north, because additional food is necessary to satisfy the high energetic requirements of overwintering in a colder place[11,40]. Northward shifts in the overwintering distributions of both Canada geese and sandhill cranes in the same region have been linked to long-term temperature trends and increasing agricultural food subsidies and, as for whooping cranes, temperatures alone cannot explain these shifts[4,22,41]. In fact, although climate is the most commonly studied driver of shortstopping[6], the first descriptions of the phenomenon attributed it to changes in land use[4]. The association between shortstopping sites and croplands in whooping cranes indicates that land use, in combination with climate change, has played an important role in driving shortstopping behaviour.

Changes in migration patterns are widespread, both geographically and taxonomically[2,4]. This reintroduced population of whooping cranes provides a unique opportunity to examine lifelong migration behaviours of individuals, but, as with any experimental approach, reintroduction does not perfectly mimic natural conditions. Future studies should investigate changes in migration patterns in other systems and populations, particularly where animals have changed long-established migration routes[15]. As future advances in tracking technology enhance our ability to monitor individuals and populations[42], these same mechanisms will be able to be explored in entirely wild populations that have changed their migration patterns.

Our results suggest that in species where behaviour depends on learning and experience, populations can rapidly change their migration patterns in association with environmental changes. Shortstopping sites were first used by groups with older, more experienced birds. The innovation by older individuals—coupled with the fact that this new behaviour resulted in a population-wide shift in winter locations—indicates that older migrants transmit their learned experience to younger animals in the population. Maintaining an age structure that includes older and experienced individuals is thus crucial for the behavioural mechanisms that allow species to adapt to global change[43]. Other studies have suggested that manipulation of population age structures can have negative evolutionary and behavioural consequences[32,44]; our paper is the first to reveal links between age structure and adaptation to global environmental change.

## Methods

**Dataset.** We used data from the location database of the eastern migratory population of whooping cranes, which tracks locations of individual birds throughout their lifetimes. The data were collected in a collaborative effort by the Whooping Crane Eastern Partnership, a public/private partnership dedicated to the reintroduction of the population. Birds in the population are uniquely identifiable for most of their lifetime via coloured leg bands and very high frequency (VHF) transmitters, which enables them to be tracked by ground vehicle and fixed-wing aircraft. In the vast majority of cases, relocations involved visual observations of birds on the ground after birds were initially located by telemetry. Some birds carried Argos satellite transmitters, which researchers used to find groups of birds (any use of trade, firm or product names is for descriptive purposes only and does not imply endorsement by the US Government); closest approaches were made with guidance from VHF telemetry. Each bird's specific location was then obtained by visual observation. In some cases, birds with non-functional transmitters were identified via leg-bands while in proximity to birds with transmitters, providing additional individual locations for the database. Only in rare exceptions were bird locations identified by telemetry but not visually confirmed. Furthermore, 11 birds released in 2011 and 2012 were fitted with global positioning system (GPS) transmitters that provided more precise locations every 4 h at night time, when they were usually roosting (and therefore stationary).

Training for the initial ultralight-guided migration consisted of early imprinting of birds on crane-costumed humans and training flights behind ultralights.

Ultralight-led migrations ended at Chassahowitzka National Wildlife Refuge on Florida's Gulf Coast from 2001 to 2007, and at St Mark's National Wildlife Refuge on the Florida panhandle from 2008 to 2015, except in 2011, when the migration ended at Wheeler National Wildlife Refuge in northern Alabama. Birds remained at the terminus of the ultralight training flight until they departed of their own accord in the following spring for the northward migration. All subsequent flights in both migratory directions were independent of the ultralights. Additional details about ultralight-led releases are provided in ref. 45.

**Identification of overwintering sites.** The monitoring protocol used to track this population (that is, VHF telemetry accompanied by visual confirmation) provides data on individual birds at high spatial and temporal resolution, making it more effective than ringing and recovery data in detecting shortstopping[46]. We used 117,223 locations of whooping cranes from October 2002 to March 2016 to identify overwintering sites, considering birds only after their first winter, that is, not considering individuals <1 year old. We considered an individual to be overwintering if it was observed in the same location (two observations <10 km apart) for at least 15 days during the winter months (November–April). These months are the dates at which birds first begin moving south and last begin moving north (Supplementary Fig. 3), and thus cover the entire overwintering period. When an individual bird was classified as having multiple overwintering sites in a single year ($N = 142$, 19% of bird-winters), we used the southernmost overwintering site to obtain a conservative estimate of the emergence of shortstopping behaviour.

These calculations resulted in 739 overwintering locations of 175 individuals. From these locations, we were able to identify overwintering sites by grouping individual overwintering locations together based on their proximity. We considered two individuals to be at the same overwintering site if their known locations were within 10 km of each other during the period they were considered to be overwintering. We repeated our analyses (described below) using two alternate methods of defining overwintering sites: one considered individuals to be at the same site if the centroids of their wintering locations were within 20 km of each other and the other considered individuals to be at the same site if the centroids of their wintering locations were within 10 km of each other. Our results were robust to these alternative methods of site identification (Supplementary Table 1).

This method identified 68 distinct overwintering sites used by the population for the winters 2003–2015. We identified the centroid of the population's three summer pen locations—Necedah National Wildlife Refuge (44.13° N, −89.95° E), White River Marsh State Wildlife Area (43.93° N, −89.10° E) and Horicon National Wildlife Refuge (43.55° N, −88.64°E)—and calculated the distance from this summering location (43.87° N, −89.23° E) to each overwintering site using an Azimuthal equidistant projection centred at the midpoint of the training route (36° N, −86° E). We identified the year in which each overwintering site was used for the first time, and then used a linear mixed-effects model to relate the distance of a new site to predictor variables that measured the migratory experience of birds, agricultural food availability and temperature change (described below). The model also included two covariates that accounted for non-independence of our data points. To account for temporal non-independence caused by the founding of multiple sites in the same year and potential longitudinal trends in site distance, we included a random intercept of the year in which the site was first used (Supplementary Data 1). Further, because sites that are spatially close together are likely to have similar environmental characteristics, we included a spatial autocovariate[47] as an additional fixed effect in the model. All analyses were performed using the lme4 (ref. 25; v. 1.1-12) and lmerTest[24] (v. 2.0-29) packages in R[48] (v. 3.2.2). We tested for collinearity between all fixed effects in the model and found all values well below the indicator threshold of $|r| > 0.70$ (ref. 49).

**Predictors.** *Age.* We identified all of the individuals that used an overwintering site in its year of first use, which allowed us to calculate characteristics of these groups. In particular, we identified the age of the oldest bird in a group, which we considered a proxy of the migratory experience of that group[17].

*Temperature.* We identified the site-specific temperature anomaly in 2000 relative to 1900, using the University of Delaware Air Temperature & Precipitation dataset[50], by calculating the difference in mean winter temperature at each site between the 1900 to 1920, and 1990 to 2010 periods. For each of these periods, we used the mean temperature in January and February, which are the months when all birds are present on their wintering grounds (Supplementary Fig. 3). This historical time period is a conservative estimate of the date of extirpation of the original migratory population of whooping cranes from eastern North America[20], so using 1900 as a reference date provides a conservative estimate of climate change since extirpation. We used the difference in temperature between these two time periods to measure temperature anomaly in 2000 relative to 1900.

*Land use.* To measure agricultural land use intensity, we used the USDA NASS Cropland Data Layer, which provides gridded cropland data at 30 m resolution, based on satellite imagery and ground surveys[51]. From this database, we extracted per cent grain cover values for 2014. We included all grains known to be food for whooping cranes, specifically corn, wheat (of all types), sorghum, millet, oats, sunflower and peanuts[23]. Values in the Cropland Data Layer reflect cover of

different crops per grid cell, from which we extracted percent cover of these grains within a 10 km radius of the center of each overwintering site.

*Spatial autocovariate.* We included a spatial autocovariate that reflected the similarity in values between nearby sites. To calculate the autocovariate for each site, we first identified all neighbours of that site, defined as all other sites within 100 km; if there were no sites within 100 km, we located the nearest neighbour. We then calculated the autocovariate using an inverse weighting scheme by distance. The value of the autocovariate ($ac_i$) for each site $i$ was:

$$ac_i = \sum_{j \in \text{neighbors}} \left( \frac{1}{distance_{i,j}} \times y_j \right)$$

where, for every site $j$ in the group of site $i$'s neighbours, $distance_{i,j}$ is the distance between sites $i$ and $j$, and $y_j$ is the value of the response variable (that is, the distance from the breeding grounds) at site $j$ (ref. 47). We scaled the autocovariate to have a mean of zero and a s.d. of 1 to place it on a similar scale as the other predictor variables in the model. After incorporating the autocovariate, there was no detectable spatial autocorrelation in the residuals of our model output at either global or local scales, which we measured by calculating Moran's $I$ at different distances (50, 200 and 1,650 km) using the correlogram function in the R package ncf[52] (v. 1.1-7).

**Individual experience.** To test whether older individuals were more experienced with the new overwintering sites than were younger migrants, we quantified experience as the minimum distance that each individual had been from a site in previous years. For each individual in a group that first used an overwintering site, we identified the closest observation of that individual to the site in any previous year or migration season. We set a threshold distance of 10 km (that is, any distance below 10 km was set to 10 km) because any observation within 10 km would have been considered a location at the same site, and log-transformed the measure. We then analysed the relationship between site familiarity and individual age using a linear mixed-effects model[24,25] with individual identity, year and overwintering site as random intercepts (Supplementary Data 2). Moran's $I$, calculated as for the main model, showed no significant spatial autocorrelation in the residuals of this model.

**Randomization test of age.** Because the mean age of the population also increased with time since the beginning of reintroduction, we performed two randomization tests to confirm that the effect of age on the distance of each site to the breeding grounds was not simply an effect of an aging population. First, for each year, we identified all sites that were used for the first time and the groups of birds that used new sites in that year ('founding groups'). We then randomly assigned one of the founding groups to each of the new sites, which, when repeated for each year in the data, produced a randomized data set that maintained the effect of the aging population in the same way as the original data set. We performed 1,000 randomizations and, for each randomization, built the same linear mixed-effects model as with the original data. This method allowed us to compare the relative significance and magnitude of the coefficients of actual and randomized data. We used a Student's *t*-test to test whether the effect of age in the randomized data was significantly different from the effect of age in the actual data.

In a second randomization test, we used the same methods but for individual-level data. For each year, we identified all individual birds that used new sites (that is, any individual belonging to a founding group), then randomly assigned each of these individuals to one of the new sites. As for the first test, we performed 1,000 randomizations, built the same linear mixed-effects model as with the original data and used a Student's *t*-test to test whether the effect of age in the randomized data was significantly different from the effect of age in the actual data.

**Social transmission of behaviour.** To determine whether shortstopping behaviour is socially transmitted, we examined whether the relationship between shortstopping and age changed over the study period (Supplementary Data 3). On the basis of the spatial distribution of sites, we considered a bird as having short stopped if it migrated <1,200 km in a given year (Supplementary Fig. 2). We built a generalized linear mixed-effects model using lme4 (ref. 25) that related this binary variable to individual age and year. The model also included a random intercept for each individual bird because individuals entered the data set multiple times.

**Data availability.** The data supporting the main findings of this study are available within the article and its Supplementary Information files (Supplementary Data 1–3).

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

## Acknowledgements

C.S.T. and T.M. were supported by the Robert Bosch Foundation. W.F.F. was supported NSF ABI 1458748. Data were supplied by the Whooping Crane Eastern Partnership (www.bringbackthecranes.org). UDel_AirT_Precip data was provided by the NOAA/OAR/ESRL PSD, Boulder, Colorado, USA, (www.esrl.noaa.gov/psd). We thank C. Hof for his comments on the text and figures and J.A. Gill for her helpful review of the manuscript.

## Author contributions

C.S.T., S.J.C. and T.M. conceived the study. C.S.T. and T.M. performed analyses with input from S.J.C., W.F.F., K.B.-G., R.B.O. and A.E.L. The manuscript was drafted by C.S.T. and T.M. with contributions from all authors.

## Additional information

**Competing financial interests**: The authors declare no competing financial interests.

**How to cite this article**: Teitelbaum, C. S. *et al.* Experience drives innovation of new migration patterns of whooping cranes in response to global change. *Nat. Commun.* 7:12793 doi: 10.1038/ncomms12793 (2016).

