## [Peer review file · Nature Communications]

Reviewers' Comments:

Reviewer #1 (Remarks to the Author)

This is a really interesting paper that reports data from a fantastic long-term study of the migratory behaviour of marked individuals - these are exactly the sort of data that are needed to identify the processes driving the continual development of migratory routes. The finding that the age of flock members is related to changes in winter site use is fascinating, and highlights the hugely important role that social information can potentially play in determining the migratory behaviour of individuals within populations. I had two concerns with the manuscript; firstly I found the over-use of the words 'innovate/innovated' throughout the MS to be very confusing and unnecessary, and I would recommend changing these throughout (in most places the word 'used' would be more appropriate). Secondly, the analyses that are presented show that flocks containing older birds use more northerly sites but they do not show the spread of information from old to young birds; given the importance placed on this social learning aspect of the study, I think that analysing and presenting the relevant data (currently only mentioned in the text) would greatly strengthen the MS. More specific comments as follows:

Page 4, line 8: the use of the word 'innovated' here seems odd and unnecessary - it would be clearer to say 'used' in place of innovated.

Page 4, line 12: again, it would be better to say that 62 sites were used independently of ultralight training.

Page 4, line 15: this would be clearer as simply 'each new wintering site', as the phrase 'newly innovated' doesn't make much sense.

Page 4, line 18: this is an example of where the word 'innovate' is particularly confusing. Does this metric refer to the oldest bird in the group that first used each new site, or in all groups that have ever used each new site?

Page 4, line 24: it would help to explain why a winter temperature anomaly could be relevant to winter distribution - eg is this likely to influence mortality rates? A direct link to a demographic or behavioural mechanism is needed to explain the inclusion of winter temperature, as climatic changes of this sort are typically highly correlated with other environmental changes.

Page 5, line 3: it would help to spell out what 'the data' refers to here, so that it is clear what spatial and temporal non-independence is of concern, and at what scales.

Page 5, line 6: again, replacing 'innovated' with 'used' would make this sentence much clearer

Page 5, line 6: remove 'innovated'

Page 5, line 11: replace 'innovated' with 'used'

Page 5, line 13: this analysis seems to assume that distance from a site reflects experience of a site? Is the presumption that closer sites are likely to have been visited by the birds at some point? If so, I think this needs some supporting evidence (eg what is the scale of movement of cranes around passage and wintering sites?).

Page 5, line 13: change 'newly innovated site' to 'new site'

Page 5, line 13: change 'innovating' to 'using'

Page 6, line 3: remove 'innovated'

Page 6, line 4: the reason for highlighting this temperature differential is not very clear, so I think this needs some consideration of the potential mechanisms through which temperatures could be linked to site use in this species - eg are these temperature differences likely to lead to substantial differences in mortality, resource availability, energetic balance etc?

Page 6, lines 6-8: I don't think the data presented here can be used to infer that having experienced individuals 'is critical for effective adaptation to rapid environmental change', so I would suggest being more circumspect here (eg having older individuals is likely to facilitate more rapid changes...)

Page 6, lines 8-9: this is the first point at which the social component of the study is really addressed, and this is key to the idea that social learning has driven these changes in winter distribution. However, the only supporting information presented is this text description of the change in distribution of young birds. I think the paper would be stronger if this was presented in the results, with associated statistical analyses and figures.

Page 6, line 16: replace 'innovation' with 'use of new'

Page 7, line 1: change 'innovating' to 'developing'

Page 7, lines 17-18: as the links between environmental changes and the observed changes in site use are correlative, I think this sentence should be reworded - the study shows that rapid changes in migratory patterns are possible but not that these are caused by environmental changes.

Page 7, line 18: replace innovation with use

Page 7, lines 20-21: as mentioned above, this population-wide shift in winter locations (and age structures) should be analysed and presented - Fig 2

Fig 2 legend, line 5: replace 'innovated' with 'used'

Fig 3 legend, lines 1-4: This figure compares distances, not site familiarity, so the summary wording doesn't seem to be right? The second sentence would also be clearer as 'Distances between new overwintering sites and previously used migration sites were shorter for older individuals'(or maybe that should be the summary sentence?)

Page 12, line 24: typo of National

Page 13, line 2: remove 'innovated'('used for the first time' is the clearest description)

Page 13, line 3: change 'an innovated site' to 'a new site'

Page 13, line 3: please explain what metric is used here to reflect 'migratory experience' (is this age of oldest bird?)

Page 13, line 4: why is year a random effect(is this year of first use)rather than a covariate (to assess temporal changes in distance from the breeding grounds)?

Page 13, line 9: presumably these age characteristics refer to all the known (ie marked) birds in each flock, but not necessarily to every individual in each flock (unless every individual is marked and continually tracked?). If so, it would be worth explaining here that the ages of the known individuals were identified, and making some comment on how the age structure of the marked

birds is likely to reflect the age structure of the overall population.

Page 14, line 18: the meaning of 'the groups that innovated all sites within each year' is not clear - does this refer only to groups recorded using a new site for the first time, or to all groups recorded in sites that had not previously been used?

Jennifer Gill
University of East Anglia

Reviewer #2 (Remarks to the Author)

The manuscript "Experience drives innovation of new migration patterns in response to global change" is a compelling paper on a relevant and timely topic of mechanisms underlying species response to patterns of global change. The authors use a unique longitudinal dataset of migration distances of individual cranes to test hypothesis about intrinsic (age) and extrinsic (land use and temperature) mechanisms that affect migration distances. Nature readers will find this paper of interest but most importantly, publication of this manuscript will help to inspire and motivate other researchers to start to think about the mechanisms that constrain or enable responses to global change - an important next step after years of publications that are describing common patterns. I thought the paper was well written, clear, concise, and a novel contribution to the field. I have a few general and specific comments that the authors may find helpful for improving their paper.

Is the advantage of shortstopping in cranes known or could it be hypothesized? For example, does shortstopping facilitate earlier arrival to the breeding grounds and does this affect reproduction? Could cranes benefit from decreased energetic expenditure by flying shorter distances?

In Figure 1 there is an example of a bird that does not shortstop, despite being several years old. Any thoughts about individuals who shortstop and individuals who do not?

I think the introduction could use a more information about a typical group size and compositions for cranes, including typical age structures. Do old birds move with old birds? What sort of variance is there in age structure?

Are there social hierarchies in cranes? I bring up the social hierarchy question because in many facultative partial migrant populations, subordinate or younger birds are forced to migrate, or migrate farther than older more dominate individuals. Could any social mechanisms be confounding your age analysis?

The term 'innovative' - seems a bit overused and is it defined until page 13

P4, Line 19: what does 'migratory performance' mean?

P7 Line 9: check out Plummer et al. 2015, Global Change Biology on this topic.

P12, Line 7: Please provide justification of selecting these months for 'winter'.

P13 Line 7: insert ' $|r| >$ ' before 0.70

P13 Line 13: Methods for temperature-I think the description of winter anomaly calculation would benefit from more details. What days were used? Why was the mean temperature used instead of minimum temperatures that tend to be more limiting in winter months? Also, it isn't clear why the authors didn't calculate year-specific anomalies when each innovation occurred?

Figure 1. It is difficult to see the size differences in symbols, perhaps consider using different shapes - like squares - to show previously used overwintering sites?

Figure 2. Great visualization of your data.

Reviewer #3 (Remarks to the Author)

This study addresses possible mechanisms behind recent shifts in overwintering locations in a migratory species, the American crane, which has shortened migratory distances over time. While these shifts have previously been documented in several migratory species, I agree the mechanisms are poorly understood. The study suggests that the involved mechanisms are both extrinsic and intrinsic. New overwintering sites are characterized by having experienced higher increases in temperatures and more extended crop cover. More controversially, the authors suggest that these sites are mainly adopted by groups containing more experienced individuals because they are more innovative in this behavioral domain. While I think these findings are potentially interesting, I also have a number of general concerns and specific issues that in my opinion require further attention.

General comments

Statistical issues:

Given the existence of well-known climatic and environmental latitudinal gradients, spatial autocorrelation can influence why new overwintering sites are associated with crop productivity and climatic anomalies. The authors acknowledge this and argue that they dealt with this possible bias by controlling for autocorrelation in the analyses. However there are too few details to know what they really did (see below). Moreover, a substantial number of analyses apparently do not incorporate spatial correlation.

In addition, the way how individual's age is related to migratory distance is problematic, in my opinion. The authors identify individuals using a novel site for first time, taking a buffer distance of 10 km, and then took the age of the oldest individual as predictor. However, this depends on group size: by chance larger groups will also have a higher probability to contain older individuals. And what if groups with very old individuals also contain very young individuals? If so, how can one distinguish which individuals are responsible of the decisions. A randomization to show how at any given year the observed proportion of older individuals is higher close to the breeding areas than expected by chance would be useful here.

The temporal issue is also important, as again acknowledged by the authors. If the population gets older over time, then any mean decrease over time in migratory distance will be spuriously associated with age as mean migratory distance has decreased over year. Including year as random effect in mixed models does not solve the problem because it just estimates a different intercept relating age and migratory distance for every year. This does not control for autocorrelation between years. I also do not see how the randomizations the authors use resolve the problem. In my opinion, what it should be demonstrated is that the proportion of older individuals does not increase over time. If it does, then it is difficult to discern whether the effect of age on migratory distance reflects experience or simply is a spurious result.

Finally, I think it would be convincing to see a longitudinal analysis showing how older birds change of overwintering areas over the years, and test whether this occurs as they age independently of the calendar year. Although a similar analyses for younger individuals is problematic, as the time series is shorter, at least it could serve to show that younger individuals do not change from overwintering site at the same time than older individuals.

Interpretation of the results:

I also have doubts about how to interpret the results. To me, the results do not necessarily suggest that experience drives innovation of new migration patterns. The same finding could merely reflect that dominant animals tend to spend the winter close to their breeding areas or that older individuals tend to do so to ensure their last chances to reproduce. This is still interesting, but should be discussed.

I also wonder how non-reproductive juveniles were considered in the analyses. These animals may disperse further as they do not need to reproduce, potentially biasing the results. In species that delay the age of first reproduction this problem could be important.

Studies focusing on a highly endangered species have additional conservation value. However, I wonder whether this is a problem for the present study as the initial overwintering locations were also novel. Thus, I would be more amazed from shifts in a species with a well-established route than in another in which the route is not yet well established. The possibility of biases in older individuals due to human-imprinting should also be discussed.

Specific issues

Abstract, line 14: changes in land use? Did the study really address changes?

Page 3, line 22: How have they been monitored? This should be advanced from the very beginning. Also, information on individuals monitored with GPS should be better exploited to show that for the studied species ringing and recovery data is efficient at detecting migratory short-stopping (Tománková et al. 2013).

Page 4, line 15: Distance from where?

Page 4, line 23: You mean in overwintering sites?

Page 4, line 25: be more specific about the meaning of "anomaly".

Page 5, line 2: covariates or predictors?

Page 5, line 3: Please clarify what you mean by spatial autocovariate and provide references. This is crucial as overwintering sites closer to the breeding areas are likely to exhibit environmental autocorrelation due to latitudinal trends in temperature (see comment above).

Page 6, line 1: The term crop is quite general and I imagine cranes do not use all types of crops. Authors should refine this analysis to make it more convincing.

Page 6, second paragraph: It should also be mentioned that behavioral shifts are observed despite migratory species generally exhibit little behavioral plasticity (Sol et al. 2005; Mettke-Hofmann 2014). Some thoughts on whether decreasing migratory distance is really an innovation should be added.

Page 13, line 17: Unclear to me what this means.

Page 14, line 1: More details are required here. Did you use lme4? How did you deal with the autocorrelation in the structure of errors?

Page 14, line 10: I wonder whether 20-10 km is too short for a bird with such flying capabilities as cranes. They probably fly much longer distances every day to get food. What is the minimum distance between overwintering sites?

Extended Data Figure 1. A more usual way to do so is to present an histogram with all the effect sizes estimated through randomizations and then indicate the position of the observed value.

Extended Data Table 1: I'm not very familiar with spatially explicit GLMM, yet it seems that the spatial autocovariate is simply added as a predictor. I think this is incorrect. Also, random effects for years should be presented in the table.

I do not want to end with a list of criticisms. I appreciate all the effort and good ideas the authors have put into this study, so I encourage them to tackle the issues I raise to produce a stronger work.

Mettke-Hofmann C (2014) Cognitive ecology: Ecological factors, life-styles, and cognition. *Wiley Interdiscip. Rev. Cogn. Sci.* 5:345-360.

Sol D, Lefebvre L, Rodríguez-Teijeiro JD (2005) Brain size, innovative propensity and migratory behaviour in temperate Palearctic birds. *Proc R Soc London B* 272:1433-41. doi: 10.1098/rspb.2005.3099

Tománková I, Reid N, Enlander I, Fox a. D (2013) Ringing and recovery data prove poor at

detecting migratory short-stopping of diving ducks associated with climate change throughout Europe. Ringing Migr 28:30-38. doi: 10.1080/03078698.2013.811184

Reviewer #1 (Remarks to the Author):

This is a really interesting paper that reports data from a fantastic long-term study of the migratory behaviour of marked individuals - these are exactly the sort of data that are needed to identify the processes driving the continual development of migratory routes. The finding that the age of flock members is related to changes in winter site use is fascinating, and highlights the hugely important role that social information can potentially play in determining the migratory behaviour of individuals within populations. I had two concerns with the manuscript; firstly I found the over-use of the words 'innovate/innovated' throughout the MS to be very confusing and unnecessary, and I would recommend changing these throughout (in most places the word 'used' would be more appropriate).

We thank the reviewer for her very positive review and helpful comments. In response to this first comment, we have removed most instances of the term “innovate” from the manuscript, including in places not noted by the reviewer. Our original intent in using the word “innovate” was to emphasize that sites were being used for the first time. Thus, depending on the context, we replaced “innovated” with “established,” “used for the first time,” or “used.” In most of these cases, using “established” provides a clearer description of the process than did “innovated.” We have retained the term “innovation” in a small number of cases, mostly where we discuss literature about animal behavior; in this field, “innovation” is a commonly-used term that refers to the appearance of novel behaviors in a population (Reader and Laland 2001) and we use this language in order to relate our results to previous studies.

Reader, S. M., and K. N. Laland. 2001. Primate innovation: Sex, age and social rank differences. *International Journal of Primatology* 22:787–805.

Secondly, the analyses that are presented show that flocks containing older birds use more northerly sites but they do not show the spread of information from old to young birds; given the importance placed on this social learning aspect of the study, I think that analysing and presenting the relevant data (currently only mentioned in the text) would greatly strengthen the MS.

This is a particularly important point that was missing from the original manuscript, and in the revised manuscript we provide a new analysis that addresses the social transmission of information. This analysis is included in the revised manuscript under the heading “social transmission of behavior.” For this new analysis, we examined the probability of shortstopping in individual birds as they aged and in the population over time. To do so, we identified whether an individual shortstopped in a given year, using the same measure as in the original manuscript (migrating less than 1200 km), then built a model that predicted the probability of shortstopping from the year and the age of an individual. If shortstopping progressed only via individual learning, we would expect the probability of shortstopping to change with individual age but not over time (Letter Figure 1a). Conversely, if shortstopping was established equally in birds of all ages, we would expect similar probabilities of shortstopping independent of age (Letter Figure 1b). However, if shortstopping were established by older individuals as a new behavior that then was adapted by the entire population, we would expect an effect of both age and year where the behavior first appears in older birds and then later in younger birds as well (Letter Figure 1c).

In our data, we found positive effects of both age and year on the probability of shortstopping, meaning that in any given year, older birds are more likely to shortstop than are younger birds, but that even younger birds are more likely to shortstop as time progressed (Letter Figure 2; Figure 4 in the main text). We present more details and full results in the manuscript (page 6, lines 13-26; page 16, lines 10-16).

Letter Figure 1: Conceptual figures of possible model outcomes predicting shortstopping behavior using age of individual birds and years. (a) If shortstopping only occurs via individual learning, we would expect the probability of shortstopping to remain constant for each age class over time and see no effect of year. (b) If shortstopping is established in the population independent of age, then we would expect birds of all ages to have an equal probability of shortstopping in any given year. In this case, only year would significantly predict shortstopping. (c) If shortstopping is transmitted socially and established by older birds, then we would expect an increase in the probability of shortstopping with age, but for all birds to begin shortstopping over time. In this case, the effects of both age and year would be significant predictors of shortstopping.

Letter Figure 2: Older birds establish shortstopping behavior. The probability of shortstopping increases with both age and year, indicating that older birds initiate shortstopping but birds of all ages learn the behavior over time. This model was based on data of all overwintering whooping cranes from 2002-2015. Curves represent predictions from a linear mixed-effects model testing the effects of age and year on shortstopping, including a random intercept of individual identity. Also shown are 95% confidence intervals of the predictions produced from 1000 bootstrap replicates. Model predictions are plotted only for age and year combinations present in the data. This figure now appears as Figure 4 in the main manuscript.

More specific comments as follows:

Page 4, line 8: the use of the word 'innovated' here seems odd and unnecessary - it would be clearer to say 'used' in place of innovated.

We have made this change.

Page 4, line 12: again, it would be better to say that 62 sites were used independently of ultralight training.

We have replaced “innovated” with “established.”

Page 4, line 15: this would be clearer as simply 'each new wintering site', as the phrase 'newly innovated' doesn't make much sense.

We have made this change.

Page 4, line 18: this is an example of where the word 'innovate' is particularly confusing. Does this metric refer to the oldest bird in the group that first used each new site, or in all groups that have ever used each new site?

We have removed the word “innovation” and edited this sentence to clarify that it refers to “the age of the oldest bird in the group that first used each overwintering site.” (page 4, lines 18-19).

Page 4, line 24: it would help to explain why a winter temperature anomaly could be relevant to winter distribution - eg is this likely to influence mortality rates? A direct link to a demographic or behavioural mechanism is needed to explain the inclusion of winter temperature, as climatic changes of this sort are typically highly correlated with other environmental changes.

We agree that the mechanistic hypothesis behind the effect of climate change on shortstopping was not clear in the original manuscript. We have edited the manuscript to clarify that climate change is likely to increase suitability of northern overwintering sites by decreasing energy expenditure during the winter, which increases body condition. We cannot hypothesize a behavioral mechanism by which this response should occur (i.e., cranes are unable to detect historic climate), but by measuring climate change at a site, we assess whether its climate is more suitable than it was historically. In places where temperatures have increased, climate change may have enabled whooping cranes to overwinter where the historic population was limited by winter temperatures. We now clarify this link between temperature and site use in two places in the manuscript. First, at the beginning of the introduction, we more directly explain why temperature should enable shortstopping (“Changes in climate and land use can create suitable new sites, where warmer temperatures decrease energy expenditure when overwintering at higher latitudes and increased food availability from land use change (e.g., land conversion to agriculture) can further offset the increased energetic requirements of overwintering in a colder place”) (page 2, line 23 - page 3, line 2). We also include this rationale when introducing our model (page 5, lines 1-6: “Changes in climate could also drive shortstopping by decreasing the amount of energy required to overwinter at each site, so we calculated site-specific winter temperature anomaly. Temperature anomaly measures the difference in average winter temperature at each overwintering site between the estimated date of extirpation of the original migratory whooping crane population (ca. 1900) and today, and thus reflects the change in climatic suitability of each site over this period.”)

Page 5, line 3: it would help to spell out what 'the data' refers to here, so that it is clear what spatial and temporal non-independence is of concern, and at what scales.

We have edited this section to be more specific about the sources of spatial and temporal non-independence. This sentence now reads, “To account for temporal non-independence caused by the founding of multiple sites in the same year and potential longitudinal trends in site distance, we included a random intercept of the year in which the site was first used. Further, because sites that are spatially close together are likely to have similar environmental characteristics, we included a

spatial autocovariate (Dormann et al. 2007) as an additional fixed effect in the model.” (page 13, lines 14-16)

Page 5, line 6: again, replacing 'innovated' with 'used' would make this sentence much clearer
We have made this change, replacing “innovated” with “were the first to use.”

Page 5, line 6: remove 'innovated'
We have made this change, replacing “innovated” with “were the first to use.”

Page 5, line 11: replace 'innovated' with 'used'
We replaced “innovated” with “established.”

Page 5, line 13: this analysis seems to assume that distance from a site reflects experience of a site? Is the presumption that closer sites are likely to have been visited by the birds at some point? If so, I think this needs some supporting evidence (eg what is the scale of movement of cranes around passage and wintering sites?).

This analysis is designed to test whether older birds are more likely to have visited new overwintering sites during past migrations than are younger birds. We selected this metric, previous distance from the site, as a proxy for site familiarity. We did so instead of testing a binary variable (i.e., previously visited site/did not previously visit site) because familiarity with an area does not necessarily require a bird to have been recorded at that exact site (e.g., the birds may have flown over the area repeatedly without landing). We have now clarified this rationale in the text: “Some individuals had previously stopped over at sites that they later established for overwintering (Fig. 1b), indicating that site familiarity may be important for establishment of new sites. In addition, whooping cranes can also recognize landscape features at relatively large spatial scales (Alerstam 1996), meaning that an individual could have knowledge of the area surrounding a new overwintering site without having directly stopped over there.” (page 5, line 25-page 6, line 1)

Alerstam, T. 1996. The geographical scale factor in orientation of migrating birds. *The Journal of experimental biology* 199:9–19.

Page 5, line 13: change 'newly innovated site' to 'new site'
We have made this change.

Page 5, line 13: change 'innovating' to 'using'
We replaced “innovating” with “first using.”

Page 6, line 3: remove 'innovated'
We have made this change.

Page 6, line 4: the reason for highlighting this temperature differential is not very clear, so I think this needs some consideration of the potential mechanisms through which temperatures could be linked to site use in this species - eg are these temperature differences likely to lead to substantial differences in mortality, resource availability, energetic balance etc?

Our additions earlier in the text clarify that that temperature change reduces energy expenditure on the wintering grounds, which results in improved body condition. We have also added a section to the discussion addressing these conclusions (see below), but we have not modified this section of the text to avoid repetition.

Page 6, lines 6-8: I don't think the data presented here can be used to infer that having experienced individuals 'is critical for effective adaptation to rapid environmental change', so

I would suggest being more circumspect here (eg having older individuals is likely to facilitate more rapid changes...)

We agree that our results do not directly address the effects of lacking older individuals in a population, so we have modified this sentence. It now reads: “Our results indicate that populations that include experienced individuals are more likely to be able to effectively and rapidly adapt behaviorally to environmental change, at least in long-lived, social species that exhibit social learning” (page 7, lines 11-13).

Page 6, lines 8-9: this is the first point at which the social component of the study is really addressed, and this is key to the idea that social learning has driven these changes in winter distribution. However, the only supporting information presented is this text description of the change in distribution of young birds. I think the paper would be stronger if this was presented in the results, with associated statistical analyses and figures.

We agree that these statistical analyses were crucial to addressing the social learning component of our study. By adding an additional analysis that examines the effects of both age and year (as described above), we show that shortstopping in young birds follows that of older birds, strengthening our conclusions about the social transmission of shortstopping.

Page 6, line 16: replace 'innovation' with 'use of new'

We have made this change.

Page 7, line 1: change 'innovating' to 'developing'

This is one case where we would prefer to retain use of the word “innovating.” Other studies cited in this section refer to “innovation” and “innovative behavior,” so by using the same language we clearly link our study to previous research. In the absence of the other uses of “innovation” in the manuscript, it should be clear in this context that “innovation” refers to a specific behavioral phenomenon. However, we would be willing to change this to “developing” if the editors prefer.

Page 7, lines 17-18: as the links between environmental changes and the observed changes in site use are correlative, I think this sentence should be reworded - the study shows that rapid changes in migratory patterns are possible but not that these are caused by environmental changes.

We have rephrased the sentence to make the relationship between shortstopping and environmental change less causal. The sentence now reads: “Our results suggest that in species where behavior depends on learning and experience, populations can rapidly change their migration patterns in association with environmental changes.” (page 9, lines 17-19)

Page 7, line 18: replace innovation with use

We rephrased this sentence to remove “innovation.”

Page 7, lines 20-21: as mentioned above, this population-wide shift in winter locations (and age structures) should be analysed and presented - Fig 2

We have now added a new analysis (presented above and in the “social transmission of behavior” section of the manuscript), which addresses this comment and strengthens our conclusions about the presence of social learning in the population.

Fig 2 legend, line 5: replace 'innovated' with 'used'

We have made this change, replacing “innovated” with “first used.”

Fig 3 legend, lines 1-4: This figure compares distances, not site familiarity, so the summary wording doesn't seem to be right? The second sentence would also be clearer as 'Distances

between new overwintering sites and previously used migration sites were shorter for older individuals'(or maybe that should be the summary sentence?)

We thank the reviewer for this suggestion and have changed the figure legend to read, “**Older birds have been closer to future overwintering sites during previous migrations.**

Distances between future overwintering sites and previously used migration sites were shorter for older individuals than for young individuals, indicating that older birds may be more familiar with the areas surrounding new overwintering sites.”

Page 12, line 24: typo of National

We have made this change.

Page 13, line 2: remove 'innovated'('used for the first time' is the clearest description)

We have made this change

Page 13, line 3: change 'an innovated site' to 'a new site'

We have made this change.

Page 13, line 3: please explain what metric is used here to reflect 'migratory experience' (is this age of oldest bird?)

We now clarify in this sentence that migratory experience is defined by maximum age (page 5, line 9).

Page 13, line 4: why is year a random effect(is this year of first use)rather than a covariate (to assess temporal changes in distance from the breeding grounds)?

We include year as a random effect to account for the fact that multiple sites were founded in the same year. Using a random effect fits an independent random intercept for each year, which accounts for between-year differences in conditions (e.g., winter temperature or snowfall). These differences in conditions are not necessarily correlated across years (e.g., a warm winter may follow a cold winter), so including year as a continuous fixed effect does not serve the same purpose. At the same time, fitting a random intercept for each year can account for longitudinal trends in migration distances between years if there is a trend in the value of the random intercept over time. We now present values for the random intercepts in the supplemental information table, as requested by Reviewer 3, which provides the reader with more information about the effect of this factor in our analysis. In addition, we clarify in the methods that year is specifically the year in which a site was first used (page 13, line 13).

As the reviewer states, migration distances have decreased in the population since the beginning of the study. We have now addressed this trend with the analysis “social transmission of shortstopping,” which measures the probability of shortstopping as a function of both age and year. Using this new analysis, we are able to show the both age and year are significant predictors of shortstopping, meaning that there are both temporal and age effects on migration distance.

Page 13, line 9: presumably these age characteristics refer to all the known (ie marked) birds in each flock, but not necessarily to every individual in each flock (unless every individual is marked and continually tracked?). If so, it would be worth explaining here that the ages of the known individuals were identified, and making some comment on how the age structure of the marked birds is likely to reflect the age structure of the overall population.

Every individual in the population was uniquely identifiable and tracking effort was equal across all ages included in our study. It is extremely unlikely that an individual was present at a site but not recorded; this would only occur in cases where an individual with a non-functioning transmitter was present at a site but not at any of the times of observation of the other members of the group. Since overwintering sites are relatively small and almost all birds have functioning transmitters, it is highly unlikely that there were undetected birds present. We have clarified in this section that we identified *all* of the individuals that used a site (page 13, line 20).

Page 14, line 18: the meaning of 'the groups that innovated all sites within each year' is not clear - does this refer only to groups recorded using a new site for the first time, or to all groups recorded in sites that had not previously been used?

We were referring only to groups using a new site for the first time. This is now clarified in the text, which reads “for each year, we identified all sites that were used for the first time and the groups of birds that used each of those new sites in that year (“founding groups”)” (page 15, lines 18-20).

Jennifer Gill
University of East Anglia

Reviewer #2 (Remarks to the Author):

The manuscript "Experience drives innovation of new migration patterns in response to global change" is a compelling paper on a relevant and timely topic of mechanisms underlying species response to patterns of global change. The authors use a unique longitudinal dataset of migration distances of individual cranes to test hypothesis about intrinsic (age) and extrinsic (land use and temperature) mechanisms that affect migration distances. Nature readers will find this paper of interest but most importantly, publication of this manuscript will help to inspire and motivate other researchers to start to think about the mechanisms that constrain or enable responses to global change - an important next step after years of publications that are describing common patterns. I thought the paper was well written, clear, concise, and a novel contribution to the field. I have a few general and specific comments that the authors may find helpful for improving their paper.

Is the advantage of shortstopping in cranes known or could it be hypothesized? For example, does shortstopping facilitate earlier arrival to the breeding grounds and does this affect reproduction? Could cranes benefit from decreased energetic expenditure by flying shorter distances?

As the reviewer suggests, there are two main potential drivers of shortstopping: decreased energy expenditure on flight and earlier arrival on the breeding grounds. We have added a section to the introduction that describes possible benefits of shortstopping: "Shortstopping can benefit migrants by decreasing the amount of energy they spend on long-distance flight (Heath et al. 2012) and facilitating earlier arrival on the breeding grounds (Bearhop et al. 2005), but these benefits require finding suitable new wintering sites closer to the breeding grounds" (page 2, lines 21-23). We have also discussed possible alternative hypotheses in the discussion, including addressing differential migration due to reproduction (page 8, lines 8-19).

Bearhop, S., W. Fiedler, R. W. Furness, S. C. Votier, S. Waldron, J. Newton, G. J. Bowen, et al. 2005. Assortative mating as a mechanism for rapid evolution of a migratory divide. *Science* 310:502–504.

Heath, J. A., K. Steenhof, and M. A. Foster. 2012. Shorter migration distances associated with higher winter temperatures suggest a mechanism for advancing nesting phenology of American kestrels *Falco sparverius*. *Journal of Avian Biology* 43:376–384.

In Figure 1 there is an example of a bird that does not shortstop, despite being several years old. Any thoughts about individuals who shortstop and individuals who do not?

We include this panel to emphasize the variability of behavior in the population that is essential for innovations to take place. Individuals that do not shortstop could do so for a variety of reasons. For instance, they may have found a particularly suitable wintering ground farther south and thus have less incentive to shortstop, or they may simply not have interacted with birds that established shortstopping sites. Other variables, such as health status or weight, could play a role in driving or limiting this behavior, but these biophysical data are not generally available for this population at the temporal resolution that would be necessary to investigate these hypotheses. We also emphasize that whooping cranes are very long-lived and this study covers only a portion of their potential lifespans, so cranes that do not shortstop during the study period may very well do so later. The rate of adoption of behavior depends on the speed at which information spreads through the population and at which new behavioral innovations occur.

I think the introduction could use a more information about a typical group size and compositions for cranes, including typical age structures. Do old birds move with old birds? What sort of variance is there in age structure?

We have added this information and now include descriptive statistics of the groups of overwintering birds in the results section, along with our analysis of social transmission of behavior. This section of the text reads: "During the study period, whooping cranes overwintered

in groups ranging from one to 22 individuals (mean 3.62 ± 3.57 birds), with age ranges up to 11 years (1.32 ± 1.86 years). Groups establishing new sites tended to be smaller (2.32 ± 1.37 birds) but still had large age ranges (2.83 ± 2.93 years)” (page 6, lines 15-18).

Are there social hierarchies in cranes? I bring up the social hierarchy question because in many facultative partial migrant populations, subordinate or younger birds are forced to migrate, or migrate farther than older more dominate individuals. Could any social mechanisms be confounding your age analysis?

There are social hierarchies in cranes; in general, older and reproductive individuals are dominant and establish winter territories separate from flocks of juveniles (Stehn 1992). However, in this case this mechanism is unlikely to play a large role in shortstopping behavior. Younger individuals do eventually shortstop and sometimes even do so in concert with older birds. Further, in the wild (western) population of whooping cranes, winter territoriality results in distances on the scale of tens of kilometers between territories (Stehn 1992), not the hundreds of kilometers associated with shortstopping. We now include this information about the presence and relative importance of population social structure in the discussion (see also: page 8, lines 13-19).

Stehn, T. V. 1992. Unusual movements and behaviors of color-banded whooping cranes during winter. *Proceedings North American Crane Workshop* 6:95–101.

The term 'innovative' - seems a bit overused and is it defined until page 13

In response to this comment and that of Reviewer 1, we have removed most instances of “innovate” from the manuscript. For the most part, we have replaced “innovated” with “used for the first time,” or “established,” which are more intuitive descriptions of the phenomenon.

P4, Line 19: what does 'migratory performance' mean?

Migratory performance is a general term that refers to the optimality of migration behavior. There are a number of different measures that can be used as proxies for migratory performance (e.g., energy expenditure, use of wind). In this case, migratory performance was measured as the directness of a migration path, i.e., how close birds travelled to a straight-line path between their summering and wintering grounds (Mueller et al. 2013; Miller et al. 2016). We provide a reference in the main manuscript that includes this definition (page 3, line 9).

Miller, T. A., R. P. Brooks, M. J. Lanzone, D. Brandes, J. Cooper, J. A. Tremblay, J. Wilhelm, et al. 2016. Limitations and mechanisms influencing the migratory performance of soaring birds. *Ibis* 158:116–134.

Mueller, T., R. B. O’Hara, S. J. Converse, R. P. Urbanek, and W. F. Fagan. 2013. Social learning of migratory performance. *Science* 341:999–1002.

P7 Line 9: check out Plummer et al. 2015, *Global Change Biology* on this topic.

We thank the reviewer for this relevant reference; we have now included it in introduction and discussion sections when discussing the interactions between changes in climate and land use.

P12, Line 7: Please provide justification of selecting these months for 'winter'.

We selected November-April as winter months based on the spatial distribution of birds over the year. No birds have arrived on their wintering grounds in October (and most remain in the breeding grounds) and almost all have arrived on the summer grounds by April. We now provide a figure that shows this distribution across the year as justification for selecting November-April as winter months (Letter Figure 3; also appears as Supplementary Figure 3).

Letter Figure 3: Monthly distribution of whooping cranes in the eastern migratory population. Winter months when all whooping cranes are on their wintering grounds are January and February; months were any cranes are on their wintering grounds are November-April.

P13 Line 7: insert '|r| >' before 0.70

We have made this change.

P13 Line 13: Methods for temperature-I think the description of winter anomaly calculation would benefit from more details. What days were used? Why was the mean temperature used instead of minimum temperatures that tend to be more limiting in winter months? Also, it isn't clear why the authors didn't calculate year-specific anomalies when each innovation occurred?

Because we were using historic as well as contemporary temperature data, we used monthly (rather than daily) data. We calculated winter temperature using the winter months when all cranes were present on their wintering grounds (January and February; Letter Figure 3) in this analysis, as described in the methods section (page 14, lines 1-7). We used mean temperature, rather than minimum temperature, in order to measure overall winter energy expenditure. Cranes have a very low lower critical temperature (Chavez-Ramirez and Wehtje 2012), so we would expect winter temperature to affect them not through extremes (i.e. the minimum temperature of the winter), but because consistently cold weather would force them to have a sustained high metabolic rate.

We calculated temperature differences over a long time period (rather than year-specific anomalies) in order to assess site-specific climate change. By doing so, we identify whether long-term changes since the extirpation of the population have enabled shortstopping, rather than assessing the effects of year-to-year weather patterns. This approach using long-term temperature anomalies is common in assessments of climate change (e.g., Rayner et al. 2003; Luterbacher et al. 2004). We have added text to the manuscript that clarifies our intent (page 5, lines 3-6: "Temperature anomaly measures the difference in average winter temperature at each overwintering site between the estimated date of extirpation of the original migratory whooping crane population (ca. 1900) and today, and thus reflects the change in climatic suitability of each site over this period") and include more details on calculation in both the main text and the methods (page 5, lines 3-6; page 13, line 24).

Chavez-Ramirez, F., and W. Wehtje. 2012. Potential impact of climate change scenarios on Whooping Crane life history. *Wetlands* 32:11–20.

Luterbacher, J., D. Dietrich, E. Xoplaki, M. Grosjean, and H. Wanner. 2004. European seasonal and annual temperature variability, trends, and extremes since 1500. *Science* 303:1499–1503.

Rayner, N. A., D. E. Parker, E. B. Horton, C. K. Folland, L. V Alexander, D. P. Rowell, E. C. Kent, et al. 2003. Global analyses of sea surface temperature, sea ice, and night marine air temperature since the late nineteenth century. *Journal of Geophysical Research: Atmospheres* 108.

Figure 1. It is difficult to see the size differences in symbols, perhaps consider using different shapes - like squares - to show previously used overwintering sites?

We now show previously-used sites as smaller triangles, which makes them easier to distinguish.

Figure 2. Great visualization of your data.

Thank you!

Reviewer #3 (Remarks to the Author):

This study addresses possible mechanisms behind recent shifts in overwintering locations in a migratory species, the American crane, which has shortened migratory distances over time. While these shifts have previously been documented in several migratory species, I agree the mechanisms are poorly understood. The study suggests that the involved mechanisms are both extrinsic and intrinsic. New overwintering sites are characterized by having experienced higher increases in temperatures and more extended crop cover. More controversially, the authors suggest that these sites are mainly adopted by groups containing more experienced individuals because they are more innovative in this behavioral domain. While I think these findings are potentially interesting, I also have a number of general concerns and specific issues that in my opinion require further attention.

General comments

Statistical issues:

Given the existence of well-known climatic and environmental latitudinal gradients, spatial autocorrelation can influence why new overwintering sites are associated with crop productivity and climatic anomalies. The authors acknowledge this and argue that they dealt with this possible bias by controlling for autocorrelation in the analyses. However there are too few details to know what they really did (see below). Moreover, a substantial number of analyses apparently do not incorporate spatial correlation.

This comment points out two gaps in the original manuscript, which we have corrected in the revised version. First, we thank the reviewer for pointing out that our exact methods for calculating the spatial autocovariate were not detailed enough. We now describe in detail how we calculated and implemented the spatial autocovariate and how we tested for spatial autocorrelation. Specifically, we include a detailed description and equation for its calculation in the Methods section (page 14, lines 15-25) and explicitly state that we tested for spatial autocorrelation in the residuals of the model using Moran's I (page 14, line 25-page 15, line 3).

Second, in the original manuscript, we had not accounted for the effects of spatial autocorrelation in our analysis of the relationship between individual age and experience (site familiarity, as presented in Figure 3 in the main text). Spatial autocorrelation could affect these results if a few specific overwintering sites had been frequently used as stopovers previously. We have now accounted for spatial autocorrelation in this analysis by introducing a random effect of site (where each overwintering site has a unique identifier). This was effective in removing spatial autocorrelation, which we measured using Moran's I as in the main analysis (page 15, lines 13-14).

In addition, the way how individual's age is related to migratory distance is problematic, in my opinion. The authors identify individuals using a novel site for first time, taking a buffer distance of 10 km, and then took the age of the oldest individual as predictor. However, this depends on group size: by chance larger groups will also have a higher probability to contain older individuals. And what if groups with very old individuals also contain very young individuals? If so, how can one distinguish which individuals are responsible of the decisions. A randomization to show how at any given year the observed proportion of older individuals is higher close to the breeding areas than expected by chance would be useful here.

This comment addresses a number of concerns. First, the reviewer suggests that group size and maximum age may be related, where by chance larger groups contain older birds. We investigated this possibility but saw no relationship between group size and the age of the birds in a group (Letter Figure 4).

Letter Figure 4: Age of the oldest bird in overwintering groups of different sizes. There is no clear relationship between group size and the age of the oldest bird in a group. Sample sizes are shown in red below each boxplot.

Second, the reviewer expresses concern that using the age of the oldest bird in a group ignores the possible decision-making of other individuals in the group. The reviewer is correct in stating that the oldest individuals may overwinter with much younger birds. Unfortunately, it is not possible to directly determine which individual (if any) is ultimately responsible for decision-making, but past studies indicate that older individuals may have more impact on group decisions than do younger birds. A previous study of this population showed that the age of the oldest bird in a group is a better predictor of migratory performance than is the age of an individual (Mueller et al. 2013), which informed our choice of the age of the oldest bird in a group as a measure of the maximum possible experience of that group. This same study also showed no effect of group size on migratory performance, despite general hypotheses that navigational accuracy should increase with group size (Simons 2004).

Further, because whooping crane groups are small during migration and overwintering (as seen in boxplot above; also see Urbanek et al. 2014), using maximum age does not mask the ages of a large number of other birds. For these reasons, we had no reason to hypothesize that group size would have a direct effect on overwintering location. However, in response to the concern that group size may drive innovation of shortstopping, we also looked at the relationship between site distance and group size and saw no trend in the relationship between site distance and group size (Letter Figure 5).

Letter Figure 5: Site distances of sites founded by groups of different sizes. Sample sizes are shown in red below each boxplot.

In addition, to address this concern, we repeated our site-based analysis from the main text, adding a predictor that measures the size of the group founding a site. In doing so, we found that group size did not have a significant effect on overwintering site distance (Letter Table 1). While it would be possible to include this analysis in the main manuscript, we would prefer to retain our original analysis so as to keep the number of predictors in our model appropriate for our sample size and to limit our predictors for which we have clear hypotheses for their effects on shortstopping.

Letter Table 1: Model results predicting overwintering site distance from a model including group size. The data used are the same as those presented in the main text, but an additional predictor (size of the innovative group) was added. This new predictor does not have a significant effect on overwintering site distance.

	Estimate	95% CI
Intercept	1614.1 ***	1485.03, 1743.18
Age of oldest bird	-42.88 **	-70.02, -15.74
Temperature change	-243.89 ***	-372.54, -115.24
Corn cover	-1690.51 ***	-2155.02, -1225.99
Group size	22.89	-13.74, 59.53
Spatial autocovariate	176.19 ***	120.43, 231.95

*** $p < 0.001$; ** $p < 0.01$; * $p < 0.05$

Last, the reviewer suggests an additional randomization showing that older individuals are closer to the breeding grounds in any given year. In addition to our group-based randomization already in the manuscript, we also performed an individual-based randomization. In our original randomization, we maintained the overwintering groups present in the original dataset but randomized the sites used by each group. As suggested by the reviewer, we have now performed a similar analysis, but randomized all individuals within a given year, creating new group compositions. In other words, any individual could use any site that was used in that year. As with the original randomization, this analysis shows that older individuals establish sites farther north than would be expected by chance (Supplementary Figure 1). We have also performed an additional analysis (see comment below) where we scaled age and migration distance within each

year, which removes the association between year and age. We saw that even using relative ages and distances, older birds still establish sites closer to the breeding grounds (Letter Table 2).

Mueller, T., R. B. O'Hara, S. J. Converse, R. P. Urbanek, and W. F. Fagan. 2013. Social learning of migratory performance. *Science* 341:999–1002.

Simons, A. M. 2004. Many wrongs: The advantage of group navigation. *Trends in Ecology and Evolution* 19:453–455.

Urbanek, R. P., E. K. Szyszkoski, and S. E. Zimorski. 2014. Winter distribution dynamics and implications to a reintroduced population of migratory whooping cranes. *Journal of Fish and Wildlife Management* 5:340–362.

The temporal issue is also important, as again acknowledged by the authors. If the population gets older over time, then any mean decrease over time in migratory distance will be spuriously associated with age as mean migratory distance has decreased over year. Including year as random effect in mixed models does not solve the problem because it just estimates a different intercept relating age and migratory distance for every year. This does not control for autocorrelation between years. I also do not see how the randomizations the authors use resolve the problem. In my opinion, what it should be demonstrated is that the proportion of older individuals does not increase over time. If it does, then it is difficult to discern whether the effect of age on migratory distance reflects experience or simply is a spurious result.

This comment addresses two concerns. First, the reviewer claims that a random effect for year does not account for temporal autocorrelation because it fits an intercept independently for each year. As we address above in response to a comment by Reviewer 1, we would like to clarify that by including year as a random effect, we are able to account for *both* within-year similarities in site distance and longitudinal trends (i.e., the temporal autocorrelation of concern here).

On one hand, we include year as a random effect to account for the fact that multiple sites were founded in the same year. Using a random effect fits an independent random intercept for each year, which accounts for between-year differences in conditions (e.g., winter temperature or snowfall). These differences in conditions are not necessarily correlated across years (e.g., a warm winter may follow a cold winter), so fitting an independent random intercept for each year is the most appropriate way to deal with this variability. At the same time, using a random intercept can account for longitudinal trends in migration distances between years if there is a trend in the value of the random intercept over time. For instance, as for a fixed effect, if there was an overall negative trend in site distance over time independent of the fixed effects in the model, the values of the random intercept would simply decrease with each year.

We now clarify the dual purpose of the random effect in the manuscript when we describe our model: “to account for temporal non-independence caused by the founding of multiple sites in the same year and potential longitudinal trends in site distance, we included a random intercept of the year in which the site was first used” (page 13, lines 11-14).

In the second part of this comment, the reviewer primarily expresses concern that, if the proportion of older individuals increases over time, it becomes difficult to determine whether age and migration distance are related, or if this trend is simply due to the simultaneous aging of the population and appearance of shortstopping. Because our study begins at the beginning of reintroduction and whooping cranes are long-lived, the proportion of older individuals does increase over time in our study. Despite this aging population, age and year are not strongly correlated in our sample because we have many age classes in all but the first few years of the study. Among all birds in the sample, the Pearson correlation coefficient of age and year is 0.45. Further, we have addressed the issue of an aging population in two ways. First, we have introduced a new analysis of shortstopping over time and show that, while individuals of all ages decrease their migration distances over the study period, older individuals do so first. This analysis is described in detail below and in the section “social transmission of shortstopping” in the manuscript (page 6, lines 13-26; page 17, lines 10-16).

Second, we have replicated our model from the original manuscript using relative distances and ages within each year. This way, we compare the relative migration distance of groups of

relative age within each year, where absolute values for age and distance remain constant over time. These scaled variables thus remove the effect of the aging population. We find that the scaled variables show the same trends as do absolute distances and ages, where, in any year, the oldest individuals establish sites the closest to the breeding grounds. Below are the results of this model.

Letter Table 2: Model results using scaled age and scaled distance. This model tested the effects of the same variables as in the main text, but used scaled distance as a response, such that the mean migration distance was zero for each year. It also uses scaled age as a predictor, such that the mean age was scaled to zero within each year.

	Estimate	95% CI
Intercept	172.48	-3.36, 348.31
Scaled maximum age	-52.4 ***	-81.96, -22.85
Temperature change	-200.4 **	-338.59, -62.21
Grain cover	-1533.67 ***	-2063.12, -1004.21
Spatial autocovariate	146.72 ***	83.99, 209.44

***p<0.001; **p<0.01; *p<0.05

Models with scaled variables are difficult to interpret and we would expect absolute years of experience, not relative experience, to be the main driver of the age effect, so we have chosen not to include this model in the revised manuscript, but it should remove any concern that our results are an artifact of an aging population.

Finally, I think it would be convincing to see a longitudinal analysis showing how older birds change of overwintering areas over the years, and test whether this occurs as they age independently of the calendar year. Although a similar analyses for younger individuals is problematic, as the time series is shorter, at least it could serve to show that younger individuals do not change from overwintering site at the same time than older individuals.

We thank the reviewer for suggesting this individual-based analysis, similar to that suggested by Reviewer 1. While it is not possible to show this phenomenon via a longitudinal analysis of individual behavior, since individual age is directly linked to calendar year, we have introduced a new analysis showing that older birds shortstop before younger birds do. If shortstopping progressed only via individual learning, we would expect the probability of shortstopping to change with individual age but for the probability of shortstopping at a given age to remain constant over time (Letter Figure 1a). Conversely, if shortstopping occurred equally in birds of all ages, we would similar probabilities of shortstopping in all age classes in a given year (Letter Figure 1b).

We found positive effects of both age and year on the probability of shortstopping, meaning that in any given year, older birds are more likely to shortstop than are younger birds, but that even young birds are more likely to shortstop as time progressed (Letter Figure 2; Figure 4 in main text). We present more details and full results in the manuscript (page 6, lines 13-26; page 16, lines 10-16).

Interpretation of the results:

I also have doubts about how to interpret the results. To me, the results do not necessarily suggest that experience drives innovation of new migration patterns. The same finding could merely reflect that dominant animals tend to spend the winter close to their breeding areas or that older individuals tend to do so to ensure their last chances to reproduce. This is still interesting, but should be discussed.

It is true that dominance hierarchies can play a role in determining overwintering behavior (Alerstam and Hedenstrom 1998), and we did not address this hypothesis in the original manuscript. While it is possible that dominance plays a role in decision-making in some species,

we emphasize that our main analysis addresses overwintering site *establishment*, not overwintering site use, and young birds shortstop as well; the shift in overwintering sites occurred in young as well as old birds, just later. In populations that show differential migration by age, older birds may overwinter farther north to increase their chances of reproduction, but in this case no individuals in this population were nearing senescence (whooping cranes can live over 20 years in the wild). In order to address these concerns, we have added a section to the manuscript that addresses how shortstopping differs from differential migration by age (page 8, lines 8-19).

Alerstam, T. and A. Hedenstrom 1998. The development of bird migration theory. *Journal of Avian Biology*. 29:343–369.

I also wonder how non-reproductive juveniles were considered in the analyses. These animals may disperse further as they do not need to reproduce, potentially biasing the results. In species that delay the age of first reproduction this problem could be important.

This is an excellent point, and in other bird populations non-reproductive juveniles do disperse farther than adults (Marques et al. 2009). However, we emphasize again that we saw a population-wide shift in overwintering locations over time, not just in birds of reproductive age. We have also addressed this point in the manuscript in the new section where we discuss differential migration (page 8, lines 13-14).

Marques, P. A. M., A. M. Costa, P. Rock, and P. E. Jorge. 2009. Age-related migration patterns in *Larus fuscus* spp. *Acta Ethologica* 12:87–92.

Studies focusing on a highly endangered species have additional conservation value. However, I wonder whether this is a problem for the present study as the initial overwintering locations were also novel. Thus, I would be more amazed from shifts in a species with a well-established route than in another in which the route is not yet well established. The possibility of biases in older individuals due to human-imprinting should also be discussed.

We thank the reviewer for bringing up this issue. As in any experimental approach, it is true that the conditions experienced by reintroduced populations do not perfectly mimic those of wild animals. In particular, the migratory route and overwintering sites of this reintroduced population were not as well established as they would be in a long-existing wild population. We have added a paragraph to the discussion about this point, emphasizing that future studies should take advantage of new tracking technology to examine these same mechanisms in natural populations, especially given that shortstopping has been described in a number of wild bird populations. We do not fully understand this last comment about imprinting at older ages, but we would like to emphasize that human imprinting in general is avoided in this population via all possible preventive measures (e.g., reduced contact with humans, crane costumes and puppets) (Urbanek et al. 2005).

Urbanek, R. P., W. J. Duff, S. R. Swengel, and L. E. a Fondow. 2005. Reintroduction techniques: post-release performance of 54 sandhill cranes (1) released into wild flocks and (2) led on migration by ultralight aircraft. *Proceedings of the North American Crane Workshop* 203–211.

Specific issues

Abstract, line 14: changes in land use? Did the study really address changes?

We thank the reviewer for bringing this point to our attention. Our study addresses food subsidies from agriculture; since agriculture is, by definition, a human land use, we consider any farmland an anthropogenic food source. However, since we use absolute values and not changes in crop cover, we agree that we do not directly measure changes in land use. We have now changed phrasing throughout the manuscript to refer to “food availability from agriculture” instead of “land use change.” In addition, we now focus on crops known to provide food to whooping cranes rather than on crop cover in general (see response to a comment below). When introducing this variable, we now say, “Corn cultivation is associated with shortstopping in sandhill cranes (Lacy et al. 2015), and whooping cranes can feed on a variety of crops, largely grains (Shields and Benham

1969), so we determined the percent grain cover within 10 km of each overwintering site as a measure of food availability from agriculture” (page 4, line 24-page 5, line 1).

Lacy, A. E., J. A. Barzen, D. M. Moore, and K. E. Norris. 2015. Changes in the number and distribution of Greater Sandhill Cranes in the Eastern Population. *Journal of Field Ornithology* 86:317–325.

Shields, R. H., and E. L. Benham. 1969. Farm crops as food supplements for whooping cranes. *Journal of Wildlife Management* 33:811–817.

Page 3, line 22: How have they been monitored? This should be advanced from the very beginning. Also, information on individuals monitored with GPS should be better exploited to show that for the studied species ringing and recovery data is efficient at detecting migratory short-stopping (Tománková et al. 2013).

The population is monitored using VHF telemetry combined with visual confirmation of birds on the ground. This method, while it does not have as regular a sampling interval as GPS data, provides much more accurate and detailed information than was previously available using ringing and recovery data. Our data has two main advantages over classic ringing data. First, the entire population is identifiable, so it is possible to track individuals and we can see changes in behavior, meaning that we will know if we are seeing a different individual or the same one at a new site. Second, for ringing data recovery rates are spatially and temporally variable and depend on external factors (e.g., hunting) (Tománková et al. 2013). In this monitoring program, there is a large intentional effort to find every individual across the year; though sampling effort is not entirely uniform, it does exist throughout the entire migratory pathway and is intentionally timed to detect as many birds as possible. Recovery rates are extremely high throughout the year (Servanty et al. 2014). We now provide this statement in the methods to emphasize how our study differs from other studies of shortstopping that have used ringing and recovery data (page 12, lines 3-6: “The monitoring protocol used to track this population (i.e., VHF telemetry accompanied by visual confirmation) provides data on individual birds at high spatial and temporal resolution, making it more effective than ringing and recovery data in detecting shortstopping”).

Servanty, S., S. J. Converse, and L. L. Bailey. 2014. Demography of a reintroduced population: Moving toward management models for an endangered species, the Whooping Crane. *Ecological Applications* 24:927–937.

Page 4, line 15: Distance from where?

We measured site distance as the distance from the centroid of the population’s breeding grounds. We have modified the text and now provide coordinates in the main text and methods (page 4, line 16; page 13, line 5).

Page 4, line 23: You mean in overwintering sites?

Yes. We have made this change (page 4, line 26).

Page 4, line 25: be more specific about the meaning of "anomaly".

A temperature anomaly is the difference in temperature between two time periods. In this case, it is the difference in temperature between the periods 1900-1920 and 1990-2010. We now provide a clearer description of this metric in the text (page 5, lines 1-6).

Page 5, line 2: covariates or predictors?

We have changed “covariates” to “predictors” throughout the manuscript.

Page 5, line 3: Please clarify what you mean by spatial autocovariate and provide references. This is crucial as overwintering sites closer to the breeding areas are likely to exhibit environmental autocorrelation due to latitudinal trends in temperature (see comment above).

We have clarified this sentence to include information about the sources of spatial autocorrelation and now provide a reference (Dormann et al. 2007) in the main text as well as in

the methods section. As noted above, we have also added more details about calculation of the spatial autocovariate in the methods section (page 14, lines 15-25).

Dormann, C. F., J. M. McPherson, M. B. Araújo, R. Bivand, J. Bolliger, G. Carl, R. G. Davies, et al. 2007. Methods to account for spatial autocorrelation in the analysis of species distributional data: a review. *Ecography* 30:609–628.

Page 6, line 1: The term crop is quite general and I imagine cranes do not use all types of crops. Authors should refine this analysis to make it more convincing.

This point is important, especially given the geographic differences in crop types across the study region. Cranes are omnivores and can eat a wide variety of grain crops but do not eat non-food crops (Shields and Benha, 1969); this is an important distinction because one main crop of the southeastern United States is cotton, which is not a food source for cranes. In contrast, grains are well-known to provide food for cranes (Prange 2012). In fact, it corn such a strong attractant for sandhill cranes that they have become a pest in cornfields and corn cultivation is likely to play a role in changes in sandhill crane distributions (Shields and Benha, 1969; Lacy et al. 2015). In response to this distinction between food-providing and non-food providing crops, we have revised our analysis to specifically focus on crops known to provide food to whooping cranes. Now, instead of calculating crop cover, we calculate grain cover at each site (details in Methods: page 14, lines 8-14). Grain cover has a stronger effect on overwintering site location than did crop cover, indicating that using this more specific metric has improved our model.

Johns, B. W., E. Woodsworth, and E. Driver. 1997. Habitat use by migrant Whooping Cranes in Saskatchewan. *Proceedings of the Seventh North American Crane Workshop* 7:123–131.

Lacy, A. E., J. A. Barzen, D. M. Moore, and K. E. Norris. 2015. Changes in the number and distribution of Greater Sandhill Cranes in the Eastern Population. *Journal of Field Ornithology* 86:317–325.

Prange, H. 2012. Reasons for changes in crane migration patterns along the West-European flyway. *Proceedings of the Cranes-Climate-People Workshop, Muraviovka Park, Russia (Vol. 28)*.

Shields, R. H., and E. L. Benham. 1969. Farm crops as food supplements for whooping cranes. *Journal of Wildlife Management* 33:811–817.

Page 6, second paragraph: It should also be mentioned that behavioral shifts are observed despite migratory species generally exhibit little behavioral plasticity (Sol et al. 2005; Mettke-Hofmann 2014). Some thoughts on whether decreasing migratory distance is really an innovation should be added.

We thank the reviewer for these interesting and relevant references. It is particularly interesting to note that both of them primarily address the differences in innovative foraging behavior between migrants and residents, where migrants seem to be less innovative in foraging because they use movement to maintain a stable food supply. It is therefore particularly fascinating that we see that a migratory species is highly innovative when it comes to movement behavior. We have now addressed this perspective in the discussion section of the manuscript, stating that: “Studies of animal cognition indicate that migratory species are less innovative than residents when it comes to foraging behavior (Mettke-Hofmann and Gwinner 2003; Sol et al. 2005), but we show that migratory species are able to innovate new movement behaviors, possibly linked to their superior long-term memory (Mettke-Hofmann and Gwinner 2003) and ability to learn from experience” (page 7, lines 17-20).

Mettke-Hofmann, C. 2014. Cognitive ecology: Ecological factors, life-styles, and cognition. *Wiley Interdisciplinary Reviews: Cognitive Science* 5:345–360.

Mettke-Hofmann, C., and E. Gwinner. 2003. Long-term memory for a life on the move.

Proceedings of the National Academy of Sciences of the United States of America 100:5863–5866.

Sol, D., L. Lefebvre, and J. D. Rodríguez-Tejjeiro. 2005. Brain size, innovative propensity and migratory behaviour in temperate Palaearctic birds. *Proceedings. Biological sciences / The Royal Society* 272:1433–41.

Page 13, line 17: Unclear to me what this means.

This historical time period is a conservative estimate of the date of extirpation of the original migratory population of whooping cranes from eastern North America. We have now rephrased the text to clarify our meaning. (page 14, lines 3-7: “This historical time period is a conservative estimate of the date of extirpation of the original migratory population of whooping cranes from eastern North America (Allen 1952), so using 1900 as a reference date provides a conservative estimate of climate change since extirpation.”)

Allen, R. P. 1952. *The Whooping Crane*. The National Audobon Society, New York.

Page 14, line 1: More details are required here. Did you use lme4? How did you dealt with the autocorrelation in the structure of errors?

We now provide a detailed description of how we calculated our spatial autocovariate and tested for spatial autocorrelation in the residuals of the model. The full model was built using lme4, as cited earlier in the manuscript, but the calculation of the autocovariate did not require use of this package. We used the ncf package to test for the presence of autocorrelation using Moran's *I*.

Page 14, line 10: I wonder whether 20-10 km is too short for a bird with such flying capabilities as cranes. They probably fly much longer distances every day to get food. What is the minimum distance between overwintering sites?

The reviewer is certainly correct to note that whooping cranes can fly long distances. However, despite this ability, previous studies have shown that during migration, whooping cranes roost less than 1 km from feeding sites (Johns et al. 1997). Though the same analysis was not been done for whooping cranes during the winter, sandhill cranes roost within 10 km of feeding sites during winter (Iverson et al. 1985) and whooping cranes establish winter territories around feeding sites (Stevenson and Griffith 1946), indicating that they are also unlikely to fly far to forage. Nevertheless, it remains true that whooping cranes are capable of flying large distances and can recognize landscape features at a distance; these abilities indicate that they may gather information at larger distances than 10 km and is why we chose to represent site familiarity as a continuous variable (i.e., previous distance from site) rather than a binary variable (e.g., has the bird visited this site before?). We now justify this variable in more detail in the text (page 5, line 22-page 6, line 1: “Some individuals had previously stopped over at sites that they later established for overwintering (Fig. 1b), but whooping cranes can also recognize landscape features at relatively large spatial scales (Alerstam 1996), meaning that an individual could have knowledge of the area surrounding a new overwintering site without having directly stopped over there”). The lower threshold of 10 km that the reviewer identified here simply indicates that we would consider a bird equally familiar with a site if it was any distance 10 km of that site's centroid.

This comment also asks about the minimum distance between sites. The minimum distance between sites in our dataset is 13 km, with a mean distance of 640 km (Letter Figure 6).

Letter Figure 6: Distribution of distances between pairs of overwintering sites used by whooping cranes during the study period.

Iverson, G. C., P. A. Vohs, T. C. Tacha, S. The, W. Management, and N. Jan. 1985. Distribution and Abundance of Sandhill Cranes Wintering in Western Texas. *Journal of Wildlife Management* 49:250–255.

Johns, B. W., E. Woodsworth, and E. Driver. 1997. Habitat use by migrant Whooping Cranes in Saskatchewan. *Proceedings of the Seventh North American Crane Workshop* 7:123–131.

Stevenson, J. O., and R. E. Griffith. 1946. Winter Life of the Whooping Crane. *The Condor* 48:160–178.

Extended Data Figure 1. A more usual way to do so is to present an histogram with all the effect sizes estimated through randomizations and then indicate the position of the observed value.

We have replaced this boxplot with a histogram of effect sizes in the revised supplementary figure.

Extended Data Table 1: I'm not very familiar with spatially explicit GLMM, yet it seems that the spatial autocovariate is simple added as a predictor. I think this is incorrect. Also, random effects for years should be presented in the table.

There are many different ways to deal with spatial autocorrelation in GLMs and GLMMs. Autocovariate regression does so by adding an additional predictor to the model: the autocovariate. The autocovariate is a weighted sum or mean of response values at nearby sites,

meaning that its effect in the model represents how much of the response at any one site is determined by the response at nearby sites (Dormann et al. 2007). So, adding the autocovariate as a predictor is the correct method of autocovariate regression. We now present the random effects in this table.

Dormann, C. F., J. M. McPherson, M. B. Araújo, R. Bivand, J. Bolliger, G. Carl, R. G. Davies, et al. 2007. Methods to account for spatial autocorrelation in the analysis of species distributional data: a review. *Ecography* 30:609–628.

I do not want to end with a list of criticisms. I appreciate all the effort and good ideas the authors have put into this study, so I encourage them to tackle the issues I raise to produce a stronger work.

Reviewers' Comments:

Reviewer #1 (Remarks to the Author)

In this revised manuscript, the authors have dealt very effectively with all of the issues raised by myself and the other referees, and I have no major comments to add (just a few minor grammatical issues listed below). The new analysis that has been added is very helpful in demonstrating the effect of age on social learning, and the randomisation tests are an important addition to the paper. I congratulate the authors on a very exciting study that should be widely cited.

Page 3, line 10: the first comma should be after 'lifetime and,' (as 'in social species' is the subclause)

Page 5, line 3: anomaly should be anomalies, in both cases (second one should be anomalies measure)

Page 5, line 5: and reflects should therefore be reflect

Page 6, line 5: I think it would be clearer to say 'more likely to be familiar with' in place of 'more experienced with'

Page 7, line 2 (and line 6): should this be 'more substantial climate change'?

Jennifer Gill
University of East Anglia

Reviewer #2 (Remarks to the Author)

I think the author's did a good job addressing reviewer comments and the manuscript has been improved.

Reviewer #3 (Remarks to the Author)

The authors have made a good job dealing with my concerns. Admittedly, whether the observed shift in migratory behaviour is due to cultural transmission is not fully demonstrated. However, this is extremely difficult to demonstrate and in any case the study provides convincing evidence in support of such a possibility. Therefore, I think this is an important contribution to the field.

We were pleased to see that the reviewers had no major comments on the revised manuscript. All of the minor textual changes suggested by Reviewer 1 are now incorporated into the manuscript; her comments and the comments of the other two reviewers are included below. The other two reviewers provided no comments that required additional revisions.

Reviewer #1 (Remarks to the Author):

In this revised manuscript, the authors have dealt very effectively with all of the issues raised by myself and the other referees, and I have no major comments to add (just a few minor grammatical issues listed below). The new analysis that has been added is very helpful in demonstrating the effect of age on social learning, and the randomisation tests are an important addition to the paper. I congratulate the authors on a very exciting study that should be widely cited.

Page 3, line 10: the first comma should be after 'lifetime and,' (as 'in social species' is the subclause)

Page 5, line 3: anomaly should be anomalies, in both cases (second one should be anomalies measure)

Page 5, line 5: and reflects should therefore be reflect

Page 6, line 5: I think it would be clearer to say 'more likely to be familiar with' in place of 'more experienced with'

Page 7, line 2 (and line 6): should this be 'more substantial climate change'?

Jennifer Gill
University of East Anglia

Reviewer #2 (Remarks to the Author):

I think the author's did a good job addressing reviewer comments and the manuscript has been improved.

Reviewer #3 (Remarks to the Author):

The authors have made a good job dealing with my concerns. Admittedly, whether the observed shift in migratory behaviour is due to cultural transmission is not fully demonstrated. However, this is extremely difficult to demonstrate and in any case the study provides convincing evidence in support of such a possibility. Therefore, I think this is an important contribution to the field.